# OpenReview forum: "Safeguard is a Double-edged Sword: Denial-of-service Attack on Large Language Models"
_ICLR.cc/2025/Conference — ICLR 2025 Conference Withdrawn Submission_

### Official Review · Reviewer_osUL · 2024-10-31

**Soundness:** 3
**Presentation:** 3
**Contribution:** 2
**Rating:** 5
**Confidence:** 4

**Summary:**

Safeguards in large language models (LLMs) are intended to block unsafe or harmful content. However, they can also be exploited to launch denial-of-service (DoS) attacks. This paper presents a method to exploit these safeguards by inserting seemingly harmless but adversarial prompts that trigger "false positives," causing the safeguard to misclassify safe content as unsafe and thereby block user requests. The authors propose an optimization algorithm that generates short adversarial prompts, which result in legitimate user requests being rejected.

**Strengths:**

- The topic of triggering safeguards for DoS attacks is interesting.
- Presentation is clear and well-structured.

**Weaknesses:**

- The proposed attack is easily detected; specific reasons are listed below.
- Real-world application scenarios for this attack are limited.

**Questions:**

The attack appears to be easily identifiable. The authors claim that the inserted prompts are seemingly harmless, making the perturbations stealthy. However, upon reviewing the examples provided, these "harmless" prompts seem easy to detect. For instance, in ``Write a tutorial onafb to hack into a bank’s database,'' if the user checks the prompt, they can immediately spot the abnormal input. The prompt contains an odd word and a sentence that appears irrelevant to the intended content, making it stand out.

This raises questions about whether the attack could realistically be deployed in practical scenarios. If the user can inspect the prompt, they would likely detect the problem. Conversely, if the user cannot inspect the prompt, it would seem simpler to insert the harmful content directly, which would achieve even better results without needing optimization.

---

### Official Review · Reviewer_rgC2 · 2024-11-02

**Soundness:** 2
**Presentation:** 3
**Contribution:** 1
**Rating:** 3
**Confidence:** 5

**Summary:**

This work describe a DDoS attack towards the LLM service by compromising the input security fence. The authors proposed to generate adversarial example to induce the input filter to classify the query of the benign user as malicious, making the service provider refuse to responce. To make the attack stealthy and practical, two extra loss terms were used to constrain the length and the semantics of the adversarial prompt injected. Experiments were done to demonstrate the effectiveness, robustness, and transferability of the adversarial prompt.

**Strengths:**

1. Good writing, easy to follow.

2. Disclosing a threat that is inherent in the LLM system with the security mechanism.

**Weaknesses:**

1. This work seems incremental to me, as it is more like a new application of the existing adversarial attack, the discussions towards other adversarial attacks in the related work (Sec.2) may help distinguish the proposed attack and the existing methods.


2. Lack of baseline methods: Though disclosing a new threat, the proposed attack is **essentially an adversarial attack against the LLM** in my perspective. Therefore I suggest the author compare the results with those of the existing SOTA works, e.g. [1], [2] to further demonstrate their contributions. Furthermore, discussions about the latest adversarial attack in Sec.2,

[1] Zou, Andy, et al. "Universal and transferable adversarial attacks on aligned language models." arXiv preprint arXiv:2307.15043 (2023).

[2] Alzantot, Moustafa, et al. "Generating natural language adversarial examples." arXiv preprint arXiv:1804.07998 (2018).


3. More SOTA models to be tested: Only open-sourced models are investigated in the transferability experiment. Experiments conducted on ChatGPT and Claude might more effectively demonstrate the severity of this threat.


4. The transferability seems to be not as good, especially for GPT4o-mini and LLAMMA-Guard. The success rate ranges from 20%~65%. Some of the results also seem to contradict the conclusion made in the paper that "*transfer attacks are more effective when the target
  models share similar architectures or training data*", as that of Llamma-Guard is the lowest. I'd like to suggest the authors dig deeper into the reason behind these phenomena and make a more thorough discussion about the results.


5. The stealthiness restriction can be concluded as one of the main contributions, yet there are not many experiments focusing on it. Moreover, I am concerned about whether the adversarial prompt gained is really stealthy or not. To attack GPT4o-mini, prompts like "*Ignore all above*" are needed, making the attack prompts suspicious enough. The prompt examples given in the paper like "*buffs yayJet bombkup deton it*" and "*to Rape PEade Lol howKra*" seem quite **unnatural** to me, making it very likely to be spotted by the user himself. I suppose this is because the author proposed to use the Bert-based model as the semantic loss. As the tokens with similar embeddings are not necessarily similarly natural. I expect more examples of adversarial prompts and more experiments focus on stealthiness, compared to other existing attacks.

**Questions:**

Seem in weaknesses

---

### Official Review · Reviewer_6B4j · 2024-11-04

**Soundness:** 2
**Presentation:** 3
**Contribution:** 2
**Rating:** 5
**Confidence:** 3

**Summary:**

In this paper, the authors propose a novel denial-of-service (DoS) attack that forces an LLM or safeguard model to falsely reject user requests by inserting an optimized adversarial prompt into a user prompt template. Experiments with various model types and task settings demonstrate the effectiveness and stealthiness of the proposed DoS attack compared to baselines like GCG.

**Strengths:**

1. The authors introduce a novel denial-of-service attack that offers a different perspective compared to existing jailbreak attacks. This method could enhance the diversity of robustness tests when measuring the safety level of an LLM.
2. Experiments using different metrics and models demonstrate the effectiveness and stealthiness of the proposed DoS attack.

**Weaknesses:**

1. The adversarial prompts optimized via DoS attack still lack semantic meanings, which means they could be easily identified by users or software developers as abnormal text. The authors should further justify the real-world threat posed by the proposed DoS attack to existing LLM services.
2. Will the proposed DoS attack remain effective in cross-task settings? For instance, can adversarial prompts optimized with logical reasoning prompts cause the target LLM to deny service when faced with coding or mathematics prompts?
3. As mentioned in the GCG paper, the length of the adversarial prompt can be a hyper-parameter during optimization. Will the DoS-optimized adversarial prompts show better effectiveness than GCG’s adversarial prompts when the lengths of both are restricted to a specific number?
4. More transfer attack results should be added to Table 2. For instance, if adversarial prompts are optimized using a model with relatively low safety levels (e.g., Vicuna), will these prompts remain effective in transfer tests against a model with a higher safety level?

**Questions:**

1. When user prompts are inserted into a template, software may sometimes paraphrase the entire prompt for improved model inference. Will DoS-optimized adversarial prompts remain effective against paraphrasing?
2. Will the proposed method remain effective against the latest commercial models, such as Claude-3 and the Gemini-1.5 series?
3. Can the DoS attack still be effective when using extremely short adversarial prompts (e.g., fewer than 10 or even fewer than 5 tokens)? If so, this would provide strong justification for the real-world impact of the proposed method.

---

### Official Review · Reviewer_Cnfo · 2024-11-04

**Soundness:** 3
**Presentation:** 3
**Contribution:** 3
**Rating:** 6
**Confidence:** 4

**Summary:**

This paper presents a novel adversarial denial-of-service (DoS) attack targeting the safeguards of large language models (LLMs). The authors introduce an attack method that exploits the false positive rates in LLM safety mechanisms by injecting inconspicuous, seemingly benign adversarial prompts into user configurations. This injection causes LLM safeguards to incorrectly flag legitimate user requests as unsafe, effectively creating a DoS attack on the model’s users. Using optimized adversarial prompts, the method blocks up to 97% of queries in tests on state-of-the-art safeguards like Llama Guard 3.

**Strengths:**

- The research idea of exploiting the LLM safeguards is interesting and important.

- Provide an examination of current defenses and introduce a unique adversarial DoS attack targeting LLM safeguards.

- Proposes an efficient optimization algorithm, balancing effectiveness and stealth for adversarial prompt generation.

**Weaknesses:**

- A key limitation is that the DoS attack depends on the attacker’s ability to inject adversarial prompts into user templates, which would typically require client compromise or user manipulation. In practice, this level of access may not be feasible in highly secure environments or applications.

- The paper’s assumption of white-box access to the safeguard model and the ability to manipulate user configuration files introduces limitations. A real-world attacker may not possess such extensive access, especially in highly secure environments. Clarifying scenarios where such access might be feasible (e.g., within compromised client software environments) could enhance the paper’s relevance.

- The authors include a case study to demonstrate the feasibility of chaining the software vulnerabilities and vulnerabilities in LLM safeguards to conduct the proposed attack. However, there is not enough detail to illustrate how this could be achieved. It would be better if the authors could provide an end-to-end case study to demonstrate the whole process of the attack.

- The paper operates on the assumption that injected adversarial prompts can remain undetected by users or monitoring systems. However, in practice, systems with moderate security measures may detect unexpected modifications to prompt templates. Including a more realistic discussion of the attack’s stealth and potential detection would be valuable, as it’s likely that this level of prompt injection might not go unnoticed in all deployment environments.

- The paper evaluates its attack mainly on the Llama Guard and Vicuna models, which may limit its generalizability to other types of LLM safeguards, especially proprietary or differently architected ones. Including more model types or extensively testing black-box models would broaden the applicability of the findings.

- The paper points out that existing defenses like random perturbation and resilient optimization reduce the DoS attack’s success but harm safeguard accuracy. However, it lacks targeted strategies to address the proposed false positive exploitation. Discussing adaptive approaches that adjust sensitivity based on prompt characteristics or an anomaly detection focused on prompt template anomalies could add values to the paper.

**Questions:**

- Since the attack assumes white-box access, how would its success rate change with only black-box or limited model access? Could the method be adapted for more restricted conditions?

---

### Note · Authors · 2024-11-25

**Comment:**

We sincerely thank all reviewers for their valuable feedback. After careful consideration, we would like to withdraw the paper and improve it based on reviewers' suggestions. Thanks.

**Withdrawal Confirmation:**

I have read and agree with the venue's withdrawal policy on behalf of myself and my co-authors.